

# Retrospective on decadal progress of the NOAA/NPS ocean noise reference station network

Samara M. Haver[1,2], Robert P. Dziak[3], Leila T. Hatch[4], Joseph Haxel[5], Christopher Kavanagh[6], Haru Matsumoto[1,7], Megan F. McKenna[8,9], Lauren Roche[1], Sofie M. Van Parijs[10], Carrie C. Wall[8,9] and Jason Gedamke[11]

[1] Cooperative Institute for Marine Ecosystem and Resources Studies, Oregon State University, Newport, OR, United States of America

[2] Department of Fisheries, Wildlife, and Conservation Sciences, Oregon State University, Corvallis, OR, United States of America

[3] NOAA Pacific Marine Environmental Laboratory, Newport, OR, United States of America

[4] NOAA Office of National Marine Sanctuaries, Silver Spring, MD, United States of America

[5] Pacific Northwest National Laboratory, Coastal Sciences Division, Sequim, WA, United States of America

[6] National Park Service, Fort Collins, CO, United States of America

[7] Living Ocean Systems, Newport, OR, United States of America

[8] Cooperative Institute for Research in Environmental Sciences, University of Colorado Boulder, Boulder, CO, United States of America

[9] NOAA National Centers for Environmental Information, Boulder, CO, United States of America

[10] NOAA Northeast Fisheries Science Center, Woods Hole, MA, United States of America

[11] Office of Science and Technology, NOAA National Marine Fisheries Service, Silver Spring, MD, United States of America

Corresponding author
Samara M. Haver,
samara.haver@oregonstate.edu

## ABSTRACT

The National Oceanic and Atmospheric Administration (NOAA), in partnership with the U.S. National Park Service (NPS), established the Ocean Noise Reference Station Network (NRS) in 2014 as a foundational component of NOAA's Ocean Noise Strategy. This long-term effort aims to characterize baseline ocean ambient sound conditions across diverse marine environments and to inform management of noise impacts on protected species and habitats within U.S. waters. The NRS is now composed of 13 autonomous passive acoustic monitoring stations strategically positioned across the U.S. Exclusive Economic Zone (EEZ), extending from Arctic regions to tropical waters in depths ranging from 33 to 4,790 m. These locations include several National Marine Sanctuaries and National Parks, such as the recently designated Chumash Heritage National Marine Sanctuary off the coast of California. Each station is equipped to continuously sample low-frequency underwater sound at five kHz, enabling the detection of anthropogenic, geophysical, and biological acoustic signals. To date the network has sampled over 72 years of calibrated acoustic data. The spatial breadth and consistent methodology of the NRS allow for comparative acoustic assessments across diverse marine ecosystems. In addition to applied research functions, the NRS has served as a platform for education and training, offering opportunities for students to develop skills for marine science and data analysis. Looking forward, the NRS project team is focused on network expansion, improved data delivery, and broader integration with collaborative scientific initiatives. NRS recordings are being archived in partnership with NOAA's National Centers for Environmental Information to enhance accessibility and long-term utility. Efforts are underway to develop standardized

metadata and summary products to accompany raw audio files, making the data more usable for a wide range of stakeholders in the ocean science community. The NRS is evolving into a fully integrated national framework for ocean sound monitoring that supports scientific inquiry, management decision-making, national security interests, and public engagement with ocean acoustic environments.

## A NATIONAL OCEAN SOUND MONITORING PROGRAM IN THE UNITED STATES

Underwater passive acoustic monitoring (PAM) technologies are increasingly being used to assess ambient sound conditions, propelled by advances in recording platform technologies to yield lower-cost instruments, longer recording durations, and more efficient analysis tools. Over the past few decades, interdisciplinary collaborations between physical and biological oceanographers, marine biologists, and ecologists have grown and expanded marine soundscape research as a valuable aspect of many PAM programs.

Recognizing the importance of sound in ocean monitoring, the Global Ocean Observing System (GOOS) is currently implementing Ocean Sound as an Essential Ocean Variable (EOV). Ocean sound is one of three cross-disciplinary EOV, and the first to primarily encompass biology and ecosystem EOVs (*Tyack et al., 2023*). Leading up to the inclusion of Ocean Sound as an EOV, an international working group, the International Quiet Ocean Experiment (IQOE) cultivated an international community of ocean soundscape researchers to examine and ultimately recommend standards for soundscape monitoring (*Boyd et al., 2011*).

Understanding the potential of soundscape monitoring to efficiently provide information about protected species and environments, the United States (US) National Oceanic and Atmospheric Administration (NOAA) published the Ocean Noise Strategy (ONS) Roadmap proposing management actions and science needed to assess underwater sound impacts to protected species and environments (*Gedamke et al., 2016*). ONS actions included establishing underwater noise monitoring programs to provide information for resource management and conservation. In response to this call, researchers at NOAA's National Marine Fisheries Service, Office of National Marine Sanctuaries, and Pacific Marine Environmental Lab collaborated with the National Park Service and academic colleagues to establish the Noise Reference Station Network (NRS; *Haver et al., 2018*). The NRS, a flagship project of the ONS, documents baseline levels, sources, and trends of low-frequency ocean ambient sound.

The NRS is internationally recognized as an endorsed project of the IQOE and now engages US and international partner scientists across universities, research institutes, federal laboratories, and private industry. This paper marks the 10-year milestone of the NRS, reflecting on progress towards ONS science, management, and outreach goals,

exemplifying the utility of standardized soundscape monitoring and the ability of NOAA research to address impacts of ocean noise on protected species and environments.

The achievement of a continuous, decade-long dataset from a coordinated national network represents a significant and exceptionally rare accomplishment in ocean science. Indeed, where such long-term records exist, they have proven invaluable for quantifying decadal-scale increases in anthropogenic noise and establishing the crucial environmental baselines needed for global ocean observing (*Andrew, Howe & Mercer, 2011*; *Howe et al., 2019*; *McDonald, Hildebrand & Wiggins, 2006*; *Merchant et al., 2022*). These long-term records are few in number, typically stemming from either massive, defense-funded military infrastructure like the US Navy's Integrated Undersea Surveillance System, or large-scale cabled observatories requiring substantial capital investment. Sustaining a distributed network of autonomous recorders like the NRS across vast geographic scales for over a decade presents persistent logistical and financial challenges, making such programs the exception rather than the rule. This rarity underscores the value of the NRS dataset through which basin-scale changes in the ocean soundscape over multiple years can be reliably detected and distinguished from short-term environmental variability.

Prior to the NRS, US ambient sound monitoring projects were not coordinated at a national level. By using consistent, calibrated passive acoustic recorders, the NRS was designed to sample underwater sound conditions concurrently in widespread locations with specific comparison goals. This standardized and coordinated monitoring has enabled analyses to compare federally protected environments and other important habitats for marine animals, as well as detect and monitor the presence of soniferous cetaceans and fish. For example, NRS hydrophones deployed in Cordell Bank National Marine Sanctuary (CBNMS), offshore of California and Buck Island Reef National Monument, in the US Virgin Islands provided the first acoustic recordings from those environments providing important information about species presence and acoustic conditions (*Haver et al., 2020*; *Haver & Nuessly, 2022*). Analysis of data collected in CBNMS revealed the presence of mysticete species in months of the year that were not previously recorded *via* conventional visual observation methods. This new information from passive acoustic monitoring contributed to lengthening a seasonal vessel speed reduction program to reduce interactions between whales and large vessels (*Office of National Marine Sanctuaries, 2023*).

NRS data have also been analyzed to evaluate the acoustic impacts of shipping vessel movement in the US Exclusive Economic Zone (EEZ), testing and applying international standards for underwater sound monitoring (*Hatch et al., 2025*; *Haver et al., 2021*). Estimating the presence of vessels in different environments is important information that can be used to estimate overlap with cetaceans that rely on sound for critical life functions. While sound source detections (*e.g.*, cetaceans, vessels, wind) are often preferred, utilizing sound levels as a proxy is more efficient and depending on resources, such as personnel and computing, can provide information faster.

NRS analysis efforts have also included broader impacts of research opportunities led by undergraduate and graduate students including student-led manuscripts and technical reports (*Haver et al., 2020*; *Munger et al., 2022*; *Pearson et al., 2023*). These projects represented opportunities for students to advance their careers in interdisciplinary

ocean sciences, while providing information about the status of protected species and environments. Principal investigators for the NRS project have also developed collaborative working relationships with researchers at other federal agencies and academic institutions, resulting in methods, data sharing tools, and proposals for new applications of NRS data.

## INSTRUMENTATION, STANDARDIZED PROCESSING, AND OPEN-ACCESS DATA

The NRS currently consists of 13 passive acoustic hydrophone moorings deployed in discrete environments throughout the EEZ. The widespread distribution of sites spans the Arctic to the tropics, including several National Marine Sanctuaries and Parks, such as the recently designated Chumash Heritage National Marine Sanctuary near the Central Coast region of California (Fig. 1). Each recorder has been routinely serviced to maintain continuous sound sampling at 10 Hz–2 kHz resulting in over 72 years of acoustic recordings.

Each NRS recorder consists of a single autonomous underwater hydrophone package housed in a titanium or composite pressure case outfitted with a calibrated omni-directional hydrophone (ITC-1032), custom frequency-dependent preamplifier, and 16-bit data acquisition system (*Fox, Matsumoto & Lau, 2001*). Depending on the deployment site depth, the recorders are either suspended in the deep-sound channel between an anchor and syntactic foam float, or secured to a bottom-mounted lander platform (see *Haver et al., 2018* for details).

Over 63 cumulative years of raw audio data (.flac) are currently hosted in the NOAA National Center for Environmental Information (NCEI) Passive Acoustic Data Archive (Fig. 2). The NCEI archive also hosts data products from analyzed NRS datasets, including calibrated one-minute hybrid-millidecade (HMD; *Martin et al., 2021*) sound level metrics. The HMD sound level metric was selected as part of a multi-year Passive Acoustic Monitoring Cyberinfrastructure "SoundCoop" Project (*National Centers for Environmental Information, 2024*). Data from two NRS sites, one in the Beaufort Sea, Alaska, and another offshore of central California, were processed for the SoundCoop project according to community standards for comparison to datasets sampled over similar conditions of time or space. The design of the NRS instrumentation, including a custom frequency-dependent-gain-pre-amplifier and multi-year sampling design, provided valuable test datasets to refine the open-source/freeware software tools utilized to quantify soundscape metrics in the SoundCoop project.

The HMD soundscape metrics calculated for the NRS in the Alaskan Arctic (Beaufort Sea) and the CBNMS were processed for the SoundCoop project using MANTA software (*Miksis-Olds et al., 2021*) and calibrated to instrument-specific hydrophone sensitivity and pre-amplifier gain. To visualize long-term and seasonal trends, the HMD metrics were averaged in approximately 90-day power spectral density (PSD) sound levels (dB re 1 $\mu Pa2/Hz$) in four seasonal bins (January to March, April to June, July to September, and October to December) and the median (L50) for each year was plotted for all years of data. 10th and 90th percentile levels were also calculated for the PSD sound levels quantified for each site.

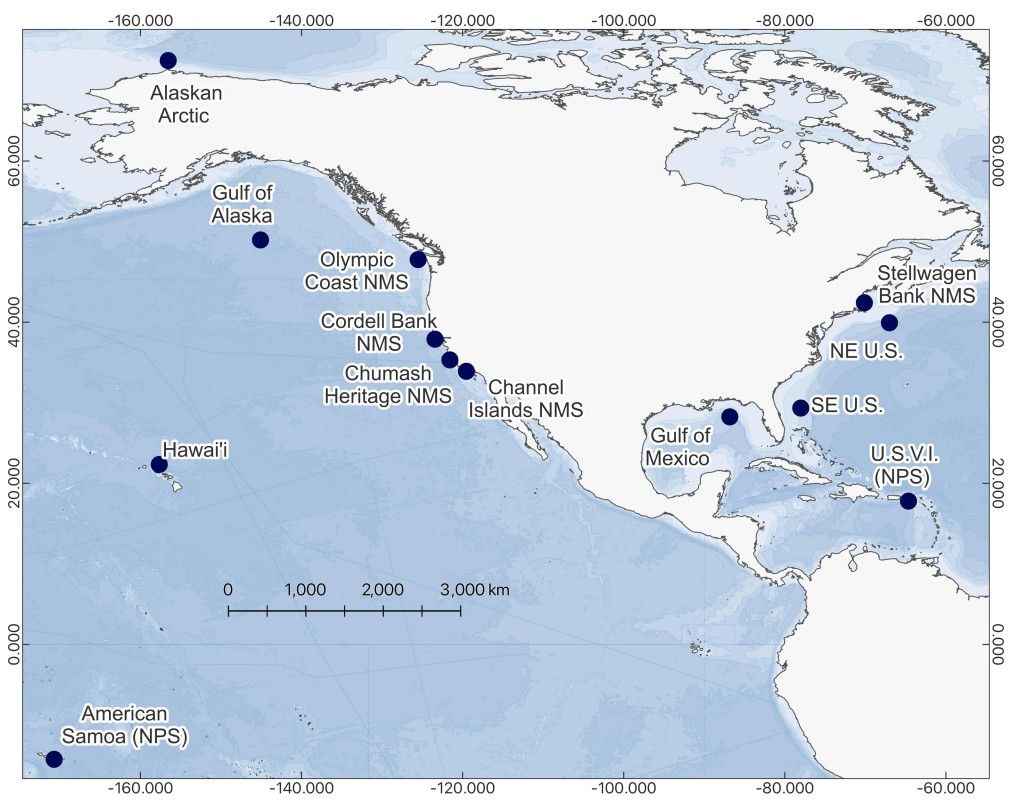

**Figure 1** **Map of 13 current NRS sites.** Alaskan Arctic (01) established 2014, Gulf of Alaska (02) established 2014, Olympic Coast National Marine Sanctuary (NMS) (03) established 2014, Hawai'i (04) established 2015, Channel Islands National Marine Sanctuary (05) established 2014, Gulf of Mexico (06) established 2014, Southeast US (07) established 2015, Northeast (NE) U.S. (08) established 2014, Stellwagen Bank National Marine Sanctuary (09) established 2014, National Park of American Samoa (10) established 2015, Cordell Bank National Marine Sanctuary (11) established 2015, US Virgin Islands (USVI) (12) established 2016, Chumash Heritage National Marine Sanctuary (13) established 2023.

To visualize the recent sound environment across the entire NRS network, spectral probability density plots (SPD; *Merchant et al., 2013*) were generated for the most recent dataset for each NRS site in Matlab (*The MathWorks Inc, 2021*). Each SPD plot quantifies the empirical probability density (EPD) of PSD sound levels to visualize the probability of occurrence of PSD sound levels in each frequency, showing sound level variation across frequency bands. Identifying common sound levels (with higher EPD values) from less common sound levels (lower EPD values) in specific frequencies can potentially indicate the presence of different sound sources and provide information about typical sound conditions in an environment. Although EPD alone may not provide enough information to distinguish between biotic and abiotic sources, probabilities of specific sound level intensity at individual frequencies across the 10 Hz to two kHz spectrum (for these data) can indicate potential sources when combined with the context the environment of the recording site and known spectral characteristics of expected sound sources in the area. To compare monitoring sites across the NRS, inclusive of the most recent years of sampling,

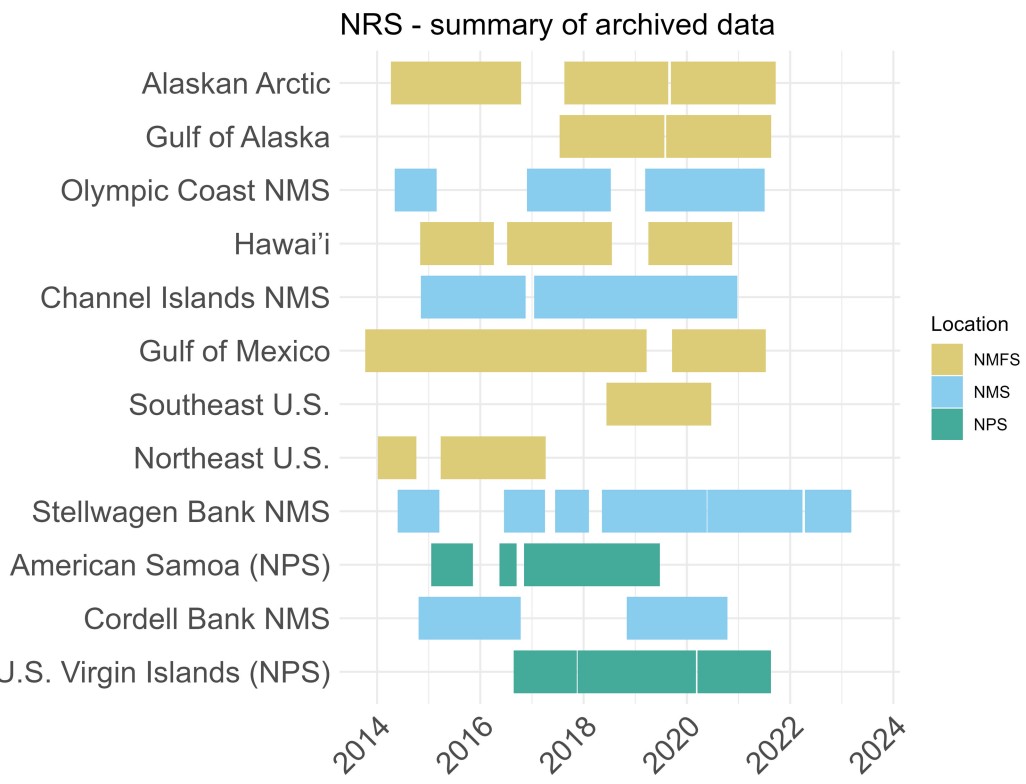

**Figure 2** **Summary of archived NRS audio data publicly available through NCEI Google Cloud Platform as of July 2025.** Datasets since 2022 are in the process of being added to the archive following instrument recovery and quality control analysis. Legend color indicates the management agency (NOAA National Marine Fisheries Service (NMFS), NOAA Office of National Marine Sanctuaries (NMS), National Park Service (NPS)) at the location of each recording site.

EPD values were summarized over time and frequency space for each entire deployment as an overall EPD value. The overall EPD value indicates the stability of sound levels over time within the given dataset which can also be utilized to compare the sound environments of different recording sites.

## ETHICS STATEMENT

All data for this paper were sampled in deep ocean environments, most greater than 500 m depth and inaccessible to humans. The three sites deployed in shallower waters were bottom mounted at depths beyond the recreational diving limit for PADI Advanced Open Water divers and can only be reached by scientific divers. Thus, it is not possible that we could have impacted the privacy of any human individuals, beyond recorded noise from the vessel they were transiting on.

# LONG-TERM TRENDS, SEASONAL PATTERNS, AND VARIABILITY IN OCEAN SOUND

Up to seven years of seasonal PSD sound level data were compared at two NRS sites, the Alaskan Arctic (Beaufort Sea) and CBNMS, to demonstrate how NRS sound level data can be used to visualize trends in soundscape conditions over time (Figs. 3 and 4).

In the Alaskan Arctic, where sea ice conditions are a primary driver of seasonal sound levels, PSD sound levels varied with the timing of maximum and minimum sea ice concentration and extent during the year (*Haver et al., 2018*). Although sea ice has a dampening effect for high-frequency sound above 500 Hz, the building and break-up of seasonally covered areas can be noisy and likely contribute to observed increases in PSD sound levels in the low-frequency range below 100 Hz. Regionally, maximum sea ice extent and concentration are typically observed in March to April, and minimum extent and concentration are typically observed in September, with break-up beginning in the summer months (*Snow & Ice Data Center, 2025*).

The annual differences in summer season PSD sound levels are likely driven by variable timing of ice break up to ice-free conditions, as well as changes in vessel movement as more of the region remains ice-free for longer. Over the 8 years of sampling, only in 2016 and 2018 was the hydrophone site still ice-covered in July; resulting in comparatively higher summer season PSD sound levels due to ice break-up in those years. The dampening effect of full sea ice coverage is observed in the Spring when median PSD sound levels are at the lowest levels above 500 Hz.

At the NRS in Cordell Bank National Marine Sanctuary, PSD sound levels are driven by whale vocalizations and vessel movement (*Haver et al., 2021*; *Haver et al., 2020*). In *Haver et al. (2020)*, species-specific analyses focused on the acoustic presence of blue whale (*Balaenoptera musculus*) 43–44 Hz b-calls, fin whale (*Balaenoptera physalus*) 20 Hz "pulse" calls, and humpback whale song and non-song vocalizations between 200 and 600 Hz (*Megaptera novaeangliae*). The data were also reviewed for the acoustic presence of gray whale (*Eschrichtius robustus*) M3 migratory calls (peak frequency of 38.1 Hz), but none were detected possibly due to animal behavior and/or the location of animals relative to the hydrophone. Multi-year comparison of seasonal conditions over time shows that PSD sound levels associated with blue and fin whale vocalizations (43–44 Hz and 20 Hz, respectively) show strong seasonality, decreasing during fall months (October–December) and increasing during winter (January–March) suggesting a seasonal shift in whale vocalization activity within the listening range of the hydrophone. Shipping vessels are a constant broadband sound source in Cordell Bank National Marine Sanctuary (*Haver et al., 2021*; *Haver et al., 2020*), but an observed decrease of PSD sound levels in Spring 2020 coincided with the onset of the COVID-19 pandemic, which was likely driven by reduced vessel activity (*Ryan et al., 2020*; *ZoBell et al., 2025*).

The NRS sustained time series of passive acoustic data sampling allows for these seasonal and annual soundscape comparisons. Continued monitoring will enable further trend analyses with comparison to target-species detection (*e.g.*, species-specific vocalizations

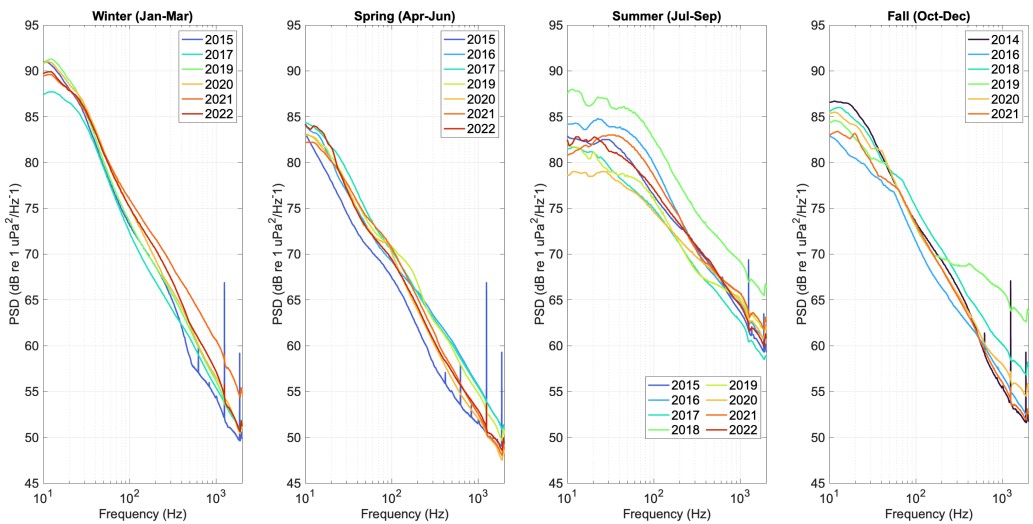

**Figure 3 Seasonal median (L50) hybrid millidecade sound levels at the Alaskan Arctic NRS site in the Western Beaufort Sea near Utquiagvik, Alaska.** Each year of data collection (2014–2022) is plotted as a different color solid line. Years without full sampling coverage for an entire three-month season were excluded from that plot.

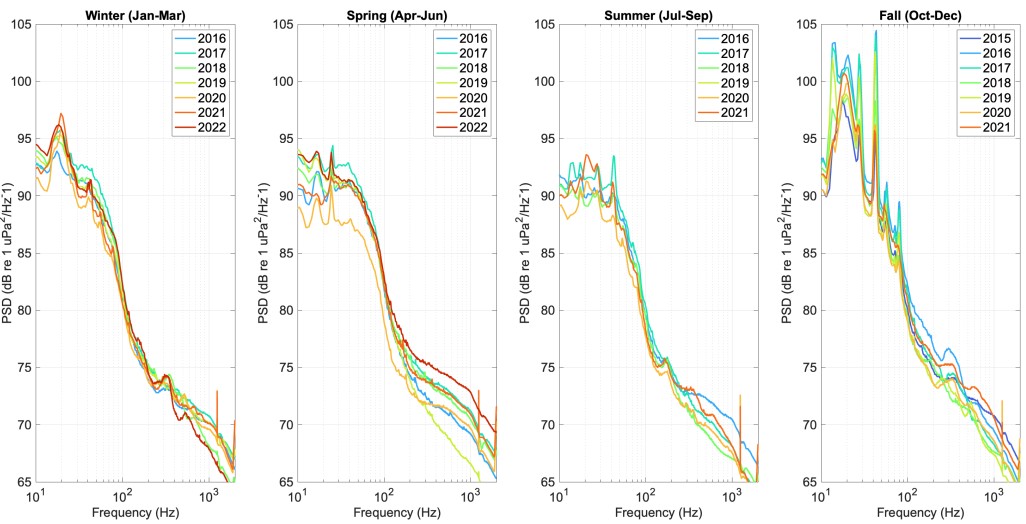

**Figure 4 Seasonal median (L50) hybrid millidecade sound levels at the Cordell Bank National Marine Sanctuary NRS site near the Bay Area region of California.** Each year of data collection (2015–2022) is plotted as a different color solid line. Years without full sampling coverage for an entire three-month season were excluded from that plot.

and anthropogenic activity) and other ocean monitoring variables (*e.g.*, wind speed, upwelling, and abundance of prey and foraging activities).

Cross-network comparison of calibrated PSD sound levels can reveal differences across ocean basins and management contexts (*e.g.*, marine protected areas). SPD plots from all

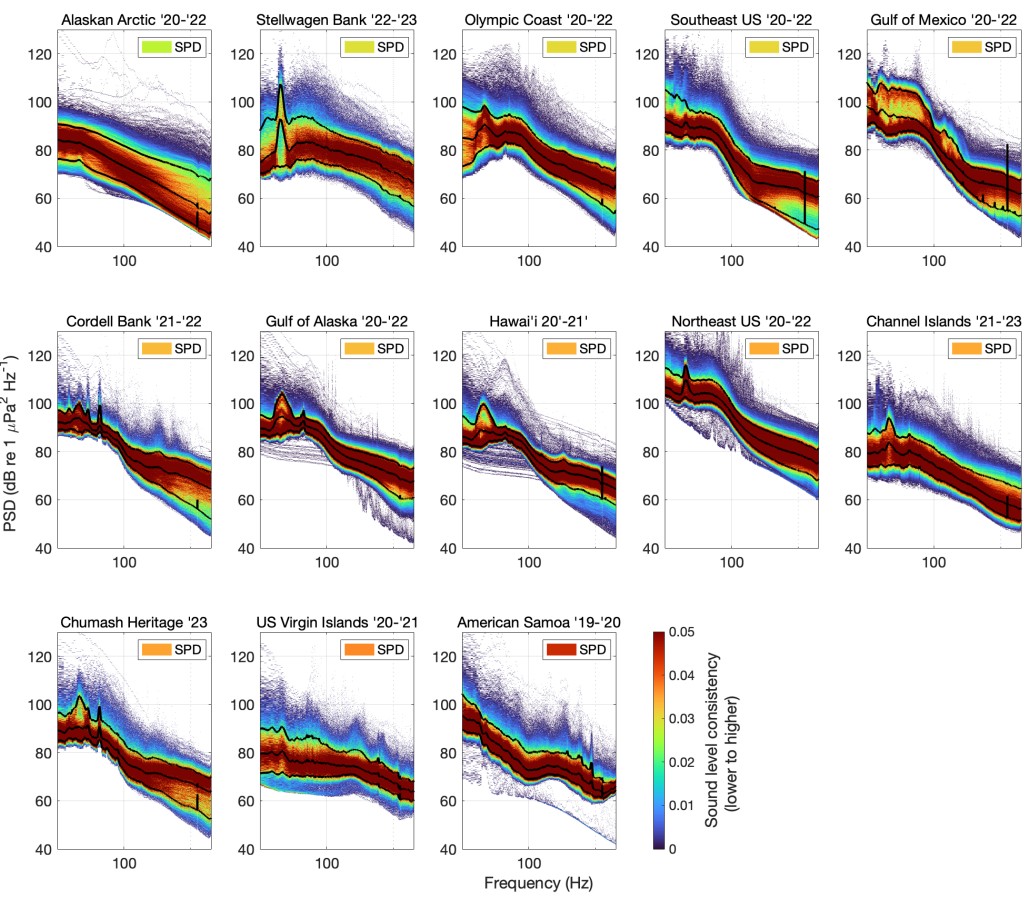

**Figure 5  Distribution of 10 Hz–2 kHz power spectral density (PSD) sound levels plotted as empirical probability density (EPD) within each frequency band.** EPD is indicated by $z$-axis color bar range of blue (lower probability) to yellow to red (higher probability). The spectral probability density (SPD) in the upper right corner of each panel indicates the overall consistency of sound levels over time (also corresponding to the $z$-axis color bar). PSD percentile levels (10th, 50th, 90th) are indicated by solid black lines.

NRS sites help to visualize how soundscape conditions differ across the network (Fig. 5). The difference between the 10th and 90th PSD percentiles levels indicate the variability of sound levels during the deployment, with a wider spread indicating more variability and a smaller difference meaning that sound levels are more consistent. At the NRS sites in the Alaska Arctic, the overall SPD is lowest among all NRS sites and the 10th and 90th percentile PSD sound levels varied by ∼20 dB across all frequencies due to the variable presence of broadband and/or multiple sound sources during seasonal shifts between ice covered and open water conditions at the NRS site. Sound levels at Stellwagen Bank NMS also varied considerably in comparison to the other sites. The median difference between 10th and 90th percentile PSD sound levels was 16.5 dB, but at 20 Hz the difference was over 34 dB. The large variability at 20 Hz is largely due to seasonal patterns in fin whale vocalizations (*De Angelis et al., 2022*; *Pittman et al., 2007*; *Watkins et al., 1987*).

In comparison, overall SPD sound levels at the American Samoa site were the least variable of NRS sites (ranging from three dB to 14 dB with a median difference of eight dB). This result is likely driven by the presence of consistent sound sources year-round coupled with lower anthropogenic activity compared to other NRS sites. PSD sound levels at the US Virgin Islands NRS were also similarly consistent during the year with a median difference of nine dB. Olympic Coast, Southeast U.S., Gulf of Mexico, Cordell Bank, Gulf of Alaska, Hawai'i, Northeast US and Chumash Heritage sound levels are less variable at 100 Hz compared to lower and higher frequencies, resulting in a lower overall SPD value. This pattern is likely related to PSD sound levels being driven by multiple sound sources, some that are primarily below 100 Hz and others that are above that threshold. For example, large shipping vessel traffic has been observed to drive sound levels in the 63 Hz and 125 Hz one-third octave bands (inclusive of frequencies 56.2–70.8 Hz and 112–141 Hz, respectively) (*Haver et al., 2021*). Blue and fin whales also commonly drive sound levels below 100 Hz as described above in reference to Fig. 4. Above 100 Hz (but below two kHz) sound sources such as wind, other cetaceans, fish, or invertebrates, and sea ice (at NRS01 only) may impact sound levels (*Haver et al., 2019*; *Haver et al., 2018*).

## THE NEXT DECADE OF MONITORING AND COLLABORATION

The first decade of the Noise Reference Station Network successfully established a durable foundation for a long-term, national-scale ocean sound monitoring program. Due to the logistical and financial challenges of ocean observing, such an achievement is significant, and is highlighted by the rarity of similarly long datasets covering such a large geographic range. As a flagship project of NOAA's Ocean Noise Strategy (*Gedamke et al., 2016*), the continuation of the NRS for the last decade demonstrates NOAA's commitment to long term passive acoustic ocean observations in order to understand the impacts of human activity on ocean habitats and marine life. Building upon this unique foundation, the NRS is now positioned to evolve beyond its first decade into a more dynamic and integral component of the national ocean science community. As demand increases for ocean sound data products, researchers have emphasized cooperative development of coordinated tools and platforms for standardized acoustic data. The NRS is an internationally recognized comprehensive sound monitoring array that plays a crucial role in this work by providing long-term, continuous, calibrated datasets. Open-access raw audio data and soundscape metric products are globally available for further analysis, including validating and benchmarking new software processing tools to enable comparison of calibrated audio datasets. Ocean soundscapes are a key area of focus, with recent projects advancing the development of standardized software tools and comparable acoustic metrics (*Wall et al., 2025*). Progress in software includes user-friendly open-source programs for noise analysis (*Marine Bioacoustics Research Collaborative, 2024*; *Martin et al., 2021*; *Miksis-Olds et al., 2021*; *Parcerisas, 2023*; *Rueda, Cline & Ryan, 2024*) and interpretation (*Sakai et al., 2025*; *Wall et al., 2025*).
In conjunction with the development of these programs, standard methods and metrics for quantifying ocean sound over frequency and time domains are being developed, as well as approaches for quantifying ambient soundscape conditions. These methodologies are under review by technical committees at the International Standards Organization (*ISO/TC 43/SC 3, 2025*), in coordination with the Acoustical Society of America, NOAA-led working groups, and other partners in the underwater sound research community.

NRS datasets are a valuable resource for the broader scientific community, as evidenced by continued demand for data and data products through the NCEI Passive Acoustic Data Archive (*Anderson et al., 2024*). In response to growing interest in calibrated ocean soundscape data, NRS will continue to archive raw data alongside standardized soundscape metric data at NCEI to facilitate data accessibility and usefulness to the broader ocean science community. Looking forward, NRS data resources will grow to include more standardized soundscape metrics (hybrid millidecade; *Martin et al., 2021*) in addition to raw audio data. These data products are formatted to a standardized network Common Data Form (netCDF) file type that combines data and metadata in a single file, simplifying public accessibility directly from NCEI open archives (*Wall et al., 2025*). The NCEI archive provides cloud-based access to its public datasets further supporting the community exploration of NRS audio data and standardized soundscape products. Free and immediate access to large volumes of data enable the scientific community to leverage scalable processing workflows, including artificial intelligence and machine learning models, to learn even more about recorded soundscapes.

Beyond continued data and data product distribution, we intend to increase the sampling rate of the network and increase data availability for collaborative work. Improvements to the autonomous passive acoustic recorders are being developed at NOAA PMEL and Oregon State University to increase the sampling rate for monitoring additional sound sources. Currently data are sampled at five kHz providing usable data in the 10 Hz to two kHz range. The upgraded system in development will be capable of sampling at least one order of magnitude greater (20 kHz) without sacrificing recording duration. Collecting data at higher frequencies will enable the NRS network to provide information about species that vocalize above two kHz such as odontocetes, pinnipeds, and fish, as well as additional man-made sound sources (*De Angelis et al., 2022*; *Kim et al., 2025*; *McKenna et al., 2021*; *Stanley et al., 2021*). These upgraded sampling capabilities will expand NRS capabilities for evaluating biodiversity and species distributions (*e.g.*, *Haver et al., 2025*), ocean noise in US waters (*e.g.*, *Dziak et al., 2025*), and potential impacts of ocean environment conditions such as marine heatwaves (*e.g.*, *Kohlman et al., 2024*).

Expanding the frequency range, duration, and number of recording sites will also provide important information about the status of ocean areas for emerging priorities for ocean energy and resource extraction. For example, proposed mining code by the International Seabed Authority suggests acoustic monitoring of biota and ambient conditions prior to and during exploration and extractive activities. Existing NRS datasets and soundscape information can provide a foundational environmental monitoring baseline, as well as inform future comparative studies to assess the potential impact of anthropogenic activities related to US energy security and independence. By providing these data and data

products *via* a public archive, NRS data can support commercial applications and needs related to technology development, national security, and marine resource management (*e.g.*, natural resource exploration and extraction, detecting illegal fishing, tracking surface and subsurface vessels).

In the next decade of the NRS project, we aim to maintain momentum to support the existing network for continuous monitoring, expand the number of sites and frequency sampling range, and increase data availability for collaborative work across NOAA offices, federal partners, and academic collaborations. In addition to ongoing research to monitor ocean noise across ecologically relevant scales, NRS datasets will be increasingly leveraged to advance NOAA Ocean Noise Strategy objectives. These include supporting assessment of noise-generating activities, engaging partners and public stakeholders on noise impacts, and fostering coordinated research. To achieve these goals, the NRS project team will prioritize data accessibility, platform improvements, and collaborative science that aligns with the research priorities of partner offices and agencies. The value of long-term datasets only increases over time, providing invaluable information about past conditions to compare in the context of current and future needs to assess ocean status and resilience to anthropogenic-driven changes.

## CONCLUSION

Over the past decade, the NRS network has continuously recorded passive acoustic data throughout US waters to document the status of ocean areas, assess species presence, and document sound environment trends in a way that was previously unattainable. The customized acoustic recorders, designed and built by NOAA researchers, are the foundation of this unique and collaborative research endeavor to combine strengths across NOAA offices and other research partners to meet data needs for a national scale understanding of marine systems.

Looking forward, the NRS will leverage its established infrastructure to foster innovation and serve a broader community of researchers, resource managers, and policymakers in the decade to come. This next phase will be defined by strategic technological advancements, network expansion, enhanced data accessibility, and deeper collaborations designed to address emerging scientific questions and pressing national priorities.

## ACKNOWLEDGEMENTS

Many collaborators and partners have contributed to the success of this research effort over the past decade. Special thanks to Jay Turnbull, Brian Kahn, Lindsey Peavey, Danielle Lipski, Jenny Waddell, Eden Zang, TK Andy Lau, Holger Klinck, David Mellinger, Chuck Anderson, Tim Rowell, Genevieve Davis, Rhett Finley, Amanda Holdman, Melissa Soldevilla, Tony Martinez, Erin Oleson, Phyllis Stabeno, Meghan Cronin, Michael Craig, Chidong Zhang, Michelle McClure, Diane Stanitski, and Jamie Shambaugh.

### Funding

This work was supported by the NOAA Ocean Acoustics Program, NOAA National Marine Fisheries Service, NOAA Pacific Marine Environmental Laboratory, NOAA Office of National Marine Sanctuaries, and the National Park Service. The funders had no role in study design, data collection and analysis, decision to publish, or preparation of the manuscript.

### Grant Disclosures

The following grant information was disclosed by the authors:
The NOAA Ocean Acoustics Program.
NOAA National Marine Fisheries Service.
NOAA Pacific Marine Environmental Laboratory.
NOAA Office of National Marine Sanctuaries.
National Park Service.

### Competing Interests

Haru Matsumoto is retired Oregon State University faculty and currently employed by Living Ocean Systems, Newport, OR, USA.

### Author Contributions

- Samara M. Haver conceived and designed the experiments, performed the experiments, analyzed the data, prepared figures and/or tables, authored or reviewed drafts of the article, and approved the final draft.
- Robert P. Dziak conceived and designed the experiments, performed the experiments, authored or reviewed drafts of the article, and approved the final draft.
- Leila T. Hatch conceived and designed the experiments, performed the experiments, authored or reviewed drafts of the article, and approved the final draft.
- Joseph Haxel conceived and designed the experiments, performed the experiments, authored or reviewed drafts of the article, and approved the final draft.
- Christopher Kavanagh conceived and designed the experiments, performed the experiments, authored or reviewed drafts of the article, and approved the final draft.
- Haru Matsumoto conceived and designed the experiments, performed the experiments, authored or reviewed drafts of the article, and approved the final draft.
- Megan F. McKenna conceived and designed the experiments, performed the experiments, authored or reviewed drafts of the article, and approved the final draft.
- Lauren Roche conceived and designed the experiments, performed the experiments, authored or reviewed drafts of the article, and approved the final draft.
- Sofie M. Van Parijs conceived and designed the experiments, performed the experiments, authored or reviewed drafts of the article, and approved the final draft.
- Carrie C. Wall conceived and designed the experiments, performed the experiments, authored or reviewed drafts of the article, and approved the final draft.

- Jason Gedamke conceived and designed the experiments, performed the experiments, authored or reviewed drafts of the article, and approved the final draft.

## Data Availability

The data is available at NOAA NCEI Passive Acoustic Data Archive:

NOAA OAR Pacific Marine Environmental Laboratory, National Marine Fisheries Service, NOS Office of National Marine Sanctuaries, and DOI NPS Natural Resource Stewardship and Science Directorate. 2014. NOAA Ocean Noise Reference Station Network Raw Passive Acoustic Data. NOAA National Centers for Environmental Information. https://doi.org/10.7289/V5M32T0D

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
