# Peer review of "Retrospective on decadal progress of the NOAA/NPS ocean noise reference station network"

_PeerJ, doi:10.7717/peerj.20382_

## Round 0.1 · original submission · Minor Revisions

· Academic Editor

Minor Revisions

An summary of NRS network progress and vision is provided. Many marine ecologists will find this paper valuable as soundscape ecology is an intriguing field and sound data is increasingly useful in unraveling complicated biological relationships. The manuscript's depth and appeal to a wider readership can be improved by some minor modifications.

Reviewer 1 ·

Basic reporting

The review paper is a retrospective review on the noise reference stations across the United States. Understanding that this is a review paper and not meant to be detailed with past findings, I think there could be some areas where more details are expanded on.

Detailed points are made below:
- Line 138 / Throughout: Instead of 600,000 hours, I would say ~68 years, since hours is quite a small unit and the number is very large. If hour is an important unit to the projects that are described, I would put the years and put the hours in parentheses.
- Line 185 - 195: When talking about sea ice extent, include a citation about where the authors got the sea-ice extent data from, or re-cite the Haver paper if the points on sea-ice are included in there.
-195 - 201: Note what whale species are vocalizing during those months, and at what frequencies are the vocalizations
- Line 220: "The large variability at 20 Hz"
- Line 223: The low variability at low-frequencies may also be likely to absence of (or low) anthropogenic activity at these sites
- Line 230: sound sources above and below 100 Hz, can you give some examples of what these sound sources may be?
- Line 251: This would be "Marine Bioacoustics Research Collaborative" instead of "Scripps Whale Acoustics Lab", or cite the Github page and website directly and put the date accessed.
- Line 260: is there data showing the "continued demand for data and data products" maybe metrics from the portal showing how many people downloaded? may need more information on how the continued demand is known / quantified
- In "the next decade" section: expansion is noted but not detailed. Some information that could be provided would be: where does this team hope to expand? What regions of the US are not yet being measured / are underrepresented by the current locations? Where will be the first areas that you target? Why? What species are not being included in the sites as they are now? What habitats are you striving to get a better representation of? Are you trying to get more pristine habitats or more degraded habitats? etc. I think having a more detailed plan about the expansion will allow readers to understand why the network is needed.

- Figures: Can you circle some of the sound sources on the spectra and note what they are (spikes showing whale song frequencies), high levels at high frequencies during weather events (?), may make the spectra figures more intuitive / accessible. Can you include some long spectrograms showing multiple years of data at these sites? I think having some spectrograms and some spectra would be good instead of just spectra figures.

Experimental design

NA

Validity of the findings

NA

Additional comments

Review paper is comprehensive, some comments on how it could be enhanced are provided in the Basic Reporting section.

Reviewer 2 ·

Basic reporting

Clearly written with sufficient references. However, I suggest the authors add some more background information. I made those comments in 'additional comment' section.

Experimental design

Methods are well explained with sufficient details but I request the authors to add some details in method section. Please see 'additional comments' section.

Validity of the findings

Although the raw data are not provided, the authors provided adequate figures to justify their findings. The conclusions are well supported by the figures .

Additional comments

The article provides an overview of the progress and vision of the NRS network. As soundscape ecology is an exciting frontier and as the usefulness of sound data in unraveling complex ecological interactions is increasing, this article will be useful for a wide range of marine ecologists as a reference point. I have some minor comments that I think will improve the depth of the manuscript and can improve the chances of appealing to wide range of audiences.

Abstract
While the abstract is informative it is very lengthy. Sometimes I lose track of what are most important goals the NOAA/NPS Ocean Noise Reference Station Network achieved since its inception. I suggest the authors shorten this section and focus on the most important aspects and achievements of this network.

Introduction
The introduction is clearly written and well structured. My only suggestion would be to add some more details about the NRS network. Although the authors provided some details about the NRS network from line 82 – 89, I think the readers will be much more benefitted to know more about NRS network (preferably in a separate paragraph.)

Methods
Line 151 – 153: Are these two stations the oldest that has the continuous data for 10 years. Among the 13 stations which are part of the NRS network, what about the other 11 stations? When were those stations included as a part of NRS network and started collecting noise data? I suggest to add a table and provide details about all the stations OR you can provide details in figure caption to avoid some redundancy.

Line 170 – 173: I suggest the authors to expand on this, preferably with an example. How do they differentiate sounds between biotic or abiotic sources (e.g. animal sound vs vessel noise). They can add a figure in that example for the readers to visually understand the difference.

Result
The structure of the this section is great and can be easily followed. I just have a minor comment (below).
Line 211: I think it’s a typo. It should be figure 5.

Discussion
This section provide adequate details about their plan that they want to carry out in the coming years. The authors also laid out the broader vision of the NRS network and discusses ways in which they plan to expand their current capabilities which I found to be an exciting undertaking.

---

## Round 0.2 · accepted · Accept

· Academic Editor

Accept

This revised version is suitable for publication in PeerJ.

Reviewer 1 ·

Basic reporting

no comment

Experimental design

no comment

Validity of the findings

no comment

Additional comments

no comment

Reviewer 2 ·

Basic reporting

The authors adequately addressed the comments.

Experimental design

The authors made the necessary changes that I think is helpful for a broader audience to understand how the study was undertaken.

Validity of the findings

Adequate changes has been made my incorporating my suggestion to validate their findings.

Additional comments

The revised manuscript can now be accepted.